# An Ontology to Standardize Research Output of Nutritional Epidemiology: From Paper-Based Standards to Linked Content

**DOI:** 10.3390/nu11061300

**Published:** 2019-06-08

**Authors:** Chen Yang, Henry Ambayo, Bernard De Baets, Patrick Kolsteren, Nattapon Thanintorn, Dana Hawwash, Jildau Bouwman, Antoon Bronselaer, Filip Pattyn, Carl Lachat

**Affiliations:** 1Department of Food Technology, Safety and Health, Ghent University, 9000 Ghent, Belgium; Chen.Yang@UGent.be (C.Y.); henryambayo@gmail.com (H.A.); Patrick.Kolsteren@UGent.be (P.K.); Dana.Hawwash@UGent.be (D.H.); 2KERMIT, Department of Data Analysis and Mathematical Modelling, Ghent University, 9000 Ghent, Belgium; Bernard.DeBaets@UGent.be; 3Department of Pathology and Anatomical Sciences, University of Missouri, Columbia, MO 65203, USA; nthanintorn@hotmail.com; 4Netherlands Organization for Applied Scientific Research, NL-2509 Zeist, The Netherlands; jildau.bouwman@tno.nl; 5Department of Telecommunications and information processing, Ghent University, 9000 Ghent, Belgium; Antoon.Bronselaer@UGent.be; 6ONTOFORCE, 9052 Ghent, Belgium; filip.pattyn@ontoforce.com

**Keywords:** ontology, nutritional epidemiology, minimal data information, data quality descriptors, study reporting guidelines, Semantic Web

## Abstract

Background: The use of linked data in the Semantic Web is a promising approach to add value to nutrition research. An ontology, which defines the logical relationships between well-defined taxonomic terms, enables linking and harmonizing research output. To enable the description of domain-specific output in nutritional epidemiology, we propose the Ontology for Nutritional Epidemiology (ONE) according to authoritative guidance for nutritional epidemiology. Methods: Firstly, a scoping review was conducted to identify existing ontology terms for reuse in ONE. Secondly, existing data standards and reporting guidelines for nutritional epidemiology were converted into an ontology. The terms used in the standards were summarized and listed separately in a taxonomic hierarchy. Thirdly, the ontologies of the nutritional epidemiologic standards, reporting guidelines, and the core concepts were gathered in ONE. Three case studies were included to illustrate potential applications: (i) annotation of existing manuscripts and data, (ii) ontology-based inference, and (iii) estimation of reporting completeness in a sample of nine manuscripts. Results: Ontologies for “food and nutrition” (*n* = 37), “disease and specific population” (*n* = 100), “data description” (*n* = 21), “research description” (*n* = 35), and “supplementary (meta) data description” (*n* = 44) were reviewed and listed. ONE consists of 339 classes: 79 new classes to describe data and 24 new classes to describe the content of manuscripts. Conclusion: ONE is a resource to automate data integration, searching, and browsing, and can be used to assess reporting completeness in nutritional epidemiology.

## 1. Introduction

Nutritional epidemiology provides evidence regarding the effects of human diets on health [1]. Unfortunately, most evidence is produced by short-term randomized trials or observational studies with small effect sizes [2]. Large-scale studies are time-consuming and demand substantial involvement of participants. Integrated analysis of shared data could increase the power of analysis and add considerable value to research [3]. However, due to the various descriptions of data and research output in nutritional epidemiology, retrieval and use of shared data is challenging. Reporting guidelines describe essential information for manuscripts and are potentially useful to standardize the description of research output [4]. 

An ontology framework developed from such guidelines enables a standardized method of data descriptions in Semantic Web [5,6]. An ontology consists of terms and their relationships to structure the description of shared data in the Semantic Web [7]. While a terminology defines the terms, an ontology defines the relationships between these terms to structure the description of shared data. Ontology terms and their relations are human-readable, but their electronic identifiers also enable computer processing such as inferencing and machine learning [8,9]. An introduction to ontology with simple examples was given by Noy and McGuinness [7]. 

Ontologies can contribute to make research output such as data, manuscripts, and study protocols findable, accessible, interoperable, and reusable (FAIR) [10]. FAIR research output is now made mandatory by research funders such as the European Commission for the establishment of a European Open Science Cloud [11]. 

The development of a virtual research infrastructure to share research output with researchers, consumers, the public, and the private sector is a promising prospect for nutrition science [12]. Despite calls since 2007 [13], progress toward an ontology for nutritional epidemiology is limited. FoodOn was developed as a taxonomy for food classification and description [14], with subsequent identifiers in Langual and FoodEx2 [15,16]. Although generic ontologies such as the Ontology for Nutritional Studies [17] and Bionutrition Ontology [18] are available, none of these enable describing nutritional epidemiologic output. 

We present the Ontology for Nutritional Epidemiology (ONE), as well as case studies to illustrate potential applications. The purpose of developing ONE was not to introduce a novel controlled vocabulary or terminology, but to define the relationships between (often existing) terms to describe nutritional epidemiology. ONE, hence, identifies relevant existing ontology terms and introduces a minimum of new terms. 

ONE has three components: (1) “descriptors for nutritional epidemiologic data”: meta-data descriptions for nutritional epidemiologic data; (2) “STROBE-nut (strengthening the reporting of observational studies in epidemiology—an extension for nutritional epidemiology) items”: quality descriptors for reporting nutritional epidemiologic studies; and (3) “nutritional epidemiologic terms”: core nutritional epidemiologic concepts. ONE is registered on Bioportal (https://bioportal.bioontology.org/ontologies/ONE), and is available on the STROBE-nut website (https://www.STROBE-nut.org) and Github (https://github.com/cyang0128/Nutritional-epidemiologic-ontologies).

The present study was conducted in the context of the European Nutritional Phenotype Assessment and Data Sharing Initiative, a collaborative effort of 16 multidisciplinary consortia from 50 research centers in nine countries, aiming to promote data sharing in nutrition.

To facilitate the reading of the article, a table of acronyms is presented (Table 1).

## 2. Materials and Methods

A scoping review of existing ontology terms provided a basis for the development of ONE. Next, ONE was developed by converting paper-based standards [4,23,24] into an ontology representation, including a separate taxonomic hierarchy of specific nutritional epidemiologic terms. Finally, ONE was applied in three case studies to illustrate its potential applications.

### 2.1. Review and Summary of Existing Ontologies for Use in Nutritional Epidemiology

As a sub-discipline of epidemiology, nutritional epidemiology analyzes the relationship between dietary intake and health [25]. As an interdisciplinary science, nutritional epidemiology also requires knowledge from other disciplines such as nutrition, food science, medicine, etc. Instead of developing a new stand-alone ontology, we firstly considered existing ontologies in epidemiology [26], as well as the relevant disciplines, and then identified missing elements for nutritional epidemiology [13]. 

On 13 April 2018, all ontologies in the three main medical ontology libraries [27,28]—OBO Foundry (http://www.obofoundry.org/) [29], BioPortal (https://bioportal.bioontology.org/) [30], and Ontology Lookup Service (https://www.ebi.ac.uk/ols/index) [31]—were reviewed by C.Y. and H.A. independently. On 26 May 2019, an update of the review was carried out to retrieve ontologies published between 13 April 2018 and 26 May 2019. Ontologies were included if their scope met part of the controlled vocabulary requirement of nutritional epidemiology, as shown in Table A1 (Appendix A). 

A pre-established data extraction spreadsheet was used to list all ontologies for review. Three review rounds were conducted. During the first review round, the full names of all the ontologies were assessed. During the second review round, the short descriptions of the ontologies on their BioPortal homepage were reviewed. If the information from the descriptions was insufficient or in case of reviewer disagreement, ontologies were included in the next review round. Finally, during the third review round, the included terms and taxonomies of the ontologies were reviewed. Disagreements were resolved through discussion until a consensus was reached. In case some ontologies were inaccessible, information for these ontologies was reviewed in relevant publications or web pages. 

The FAIR principles provide essential guidance to search and integrate data at the individual and meta-level. The required types of controlled vocabulary to achieve FAIR principles in nutritional epidemiology were summarized (Table A1, Appendix A), and the ontologies were classified accordingly. A quality assessment of the selected ontologies was conducted using the modules by Burton-Jones et al. [32]. Minor changes were made to present the quality of multiple medical ontologies. On 16 May 2018, statistics were collected through BioPortal (https://bioportal.bioontology.org/), Agroportal (http://agroportal.lirmm.fr/), and Ontobee (http://www.ontobee.org/).

### 2.2. Development of ONE

The ontology is represented in the resource description framework (RDF) format [33] and edited using the default text editor of Microsoft Windows 7. A quality assessment of ONE was conducted as proposed by Burton-Jones, Storey, Sugumaran, and Ahluwalia [32]. The relevance, authority, and history module were not assessed, however, as they require data collection after publishing the ontology.

#### 2.2.1. Existing Data Standards in Nutritional Epidemiology

The terms of two existing standards for nutrition research (i.e., minimal meta-data descriptors [23] and data quality descriptors [24]) were represented in ONE. The ontology terms were grouped as “descriptors for nutritional epidemiologic data”.

In case certain terms were found in more than one ontology, the term with the definition that best described the intended term was selected by a domain expert. When no exact terms were found in the selected ontologies, a synonym term was obtained from a domain expert if the definition was suitable.

However, if the exact term or the synonym could not be retrieved from existing ontologies, a new electronic identifier was attributed: (1) for terms only used in nutritional epidemiologic research, the identifier “one:nexxxxx” (xxxxx = five digits) was assigned, where “one” represents “ontology for nutritional epidemiology”, and “ne” represents “only used in nutritional epidemiology” (e.g., identifier for “dietary assessment administration”: one:ne00057); (2) for other terms that can also be used in other subjects, identifier “one:Txxxxx” (xxxxxx = five digits) was assigned, where “one” represents “ontology for nutritional epidemiology” and “T” represents “temporary” (e.g., identifier for “food composition table”: one:T00027). Terms indicated with “T” should, hence, be developed in their corresponding domain ontology. The list of temporary terms will be reviewed on a regular basis and updated where needed.

#### 2.2.2. Reporting Guidelines in Nutritional Epidemiology 

The “strengthening the reporting of observational studies in epidemiology” (STROBE) reporting guidelines for nutritional epidemiology [4] were used as the basis to develop the ontology for reporting of nutritional epidemiology. The collection of ontology terms is allocated under the term “STROBE-nut items” in ONE. For the STROBE-nut reporting items (e.g., title, abstract, etc.), electronic identifier “one:reportxxxxx” (xxxxx = five digits) was given, where “one” represents “ontology for nutritional epidemiology”, and “report” represents “reporting items” (e.g., identifier for “title”: one:report00001); for the STROBE-nut recommendations, identifier “one:report/nut-x” (x = one digit) was assigned, where “one” represents “ontology for nutritional epidemiology”, and “report/nut-x” represents “the STROBE-nut recommendations for reporting on items” (e.g., identifier for “STROBE-nut recommendation 1”: one:report/nut-1).

#### 2.2.3. Nutritional Epidemiologic Terms

The term “nutritional epidemiologic terms” (electronic identifier: one:terms) was used to group the specific nutritional epidemiologic terms summarized from the standard descriptions during the previous steps. The taxonomy presents terms to describe the core concepts, study design, and data measurement characteristics of nutritional epidemiology. However, those terms do not cover generic information to report research, such as study name, study duration, study area, etc. Terms used for generic study information, however, are considered part of the minimal data requirements and quality descriptors, and were, hence, mainly retrieved from other existing ontologies.

### 2.3. Applications of ONE 

ONE was applied in three case studies to illustrate its potential applications: (i) study annotation and term query, (ii) ontology-based inference, and (iii) estimation of reporting completeness in a sample of nine manuscripts. 

Firstly, an existing manuscript [34] and one of its corresponding datasets were annotated manually using ONE terms (Syntax available on Bioportal). Terms from other ontologies were also used to annotate nutrition information that was not related to nutrition (e.g., geography, season, etc.).

Secondly, the potential ontology-based inference was described. Inference on the basis of the taxonomy of terms can significantly improve the quality of data search and integration. Three terms used to annotate the manuscripts were selected for this case study [34]. By showing partial taxonomic hierarchies of the three terms, we explained how to infer unknown information from available information.

Thirdly, an assessment of reporting completeness was conducted using ONE, similar to the ontology-based meta-analyses by Kupershmidt et al. [35] and Ramaprasad and Syn [36]. A convenient sample of nine published manuscripts [37,38,39,40,41,42,43,44,45] was manually annotated using STROBE-nut terms of ONE for this purpose. By querying the electronic identifiers of STROBE-nut terms, the reporting frequencies of STROBE-nut terms were obtained. The hierarchies of STROBE-nut terms and one annotated manuscript were compared to illustrate where STROBE-nut terms were reported in the manuscript. 

## 3. Results

### 3.1. Review and Summary of Existing Ontology Vocabulary for Use in Nutritional Epidemiology

In total, 1146 ontologies were retrieved, of which 237 were selected and classified according to their scope (Figure 1). As shown in Table A2 (Appendix A), 158 ontologies were selected to annotate data (33 ontologies for food/dietary agricultural products, four ontologies for nutrients/chemical compounds, 100 ontologies for disease and specific population (e.g., student health record ontology), and 21 ontologies for data management), and 35 were selected for metadata annotation (35 ontologies for research terminology and no ontology for metadata representation). There were also 44 ontologies to describe supplementary (meta) data (e.g., ethical issues, demographics, fundamental ontology knowledge frameworks, etc.). Among the ontologies found, no ontology was developed as a frame (e.g., guidance and guidelines) to present meta-data in nutritional epidemiologic information.

The quality assessment (Figure A1a) shows that 14% of the selected ontologies had less than 100 terms. Most of the selected ontologies (65%) had 101–10,000 terms, while 15% of the selected ontologies had more than 10,000 terms. The richness module (Figure A1b) shows that 15% of the selected ontologies had no properties, 23% of the selected ontologies had 1–10 properties, and 55% of the selected ontologies had more than 10 properties, including 14% of the selected ontologies with over 100 properties. Figure A1c,d indicate that 25% of the terms had no definitions, and 94% of the selected ontologies were not peer-reviewed. The lawfulness module (Figure A1e), authority module (Figure A1f), and history module (Figure A1g) represent the practicality of the selected ontologies. Only 2% of the selected ontologies were inaccessible due to error ontology files (Figure A1e). Only 8% of the selected ontologies were not mapped, while 20% of the selected ontologies were made of more than 300 mapped ontologies (Figure A1f). Less than half (47%) of the selected ontologies were visited less than 10 times per month (Figure A1g).

### 3.2. Development of ONE

The structure of ONE is shown in Figure 2, and a quality description is included in Table A3 (Appendix A). ONE consists of 339 classes. It reuses classes from 22 existing ontologies, where the main referred medical ontologies are NCIT (National Cancer Institute Thesaurus, 43 classes) and MeSH (Medical Subject Headings, 33 classes). ONE proposes 79 new classes to describe nutrition data and 24 new classes to describe the content of manuscript.

The electronic identifiers of terms are written after the corresponding terms. The electronic identifiers (e.g., NCIT:C94729) consist of two parts: (1) an ontology acronym (e.g., “NCIT” is the acronym of “ontology for National Cancer Institute Thesaurus”), and (2) a code of the term in the ontology (e.g., C94729 is code of “season” in NCIT ontology).

#### 3.2.1. Existing Data Standards in Nutritional Epidemiology

The main taxonomies of the minimal data requirements and data quality descriptors are shown in Figure 3 and Figure 4, respectively. The collection of ontology terms is reported in Table A8 and Table A9 (Appendix A), respectively. Recommendations for generic terms that could not be found in existing ontologies of other subjects are indicated as footnotes of Table A8 and Table A9 (Appendix A). 

#### 3.2.2. STROBE-Nut Reporting Guidelines in Nutritional Epidemiology

For the collection of ontology terms for STROBE-nut reporting guidelines, the STROBE reporting items (e.g., title, abstract, etc.) were used as a taxonomic hierarchy of terms. The specific STROBE-nut recommendations were arranged under their corresponding STROBE reporting items (Figure 5). 

#### 3.2.3. Nutritional Epidemiologic Terms

Nutritional epidemiologic terms and their taxonomic hierarchy are shown in Table 2. All the terms were arranged according to the relevant descriptors listed in the minimal data requirements [23] and data quality descriptors [24]. The terms at the first hierarchy level are the descriptors (i.e., nutritional epidemiologic terms), while the terms at the second and third hierarchy levels are the options of descriptors (i.e., terms with more specific descriptions used for specific conditions). The taxonomic hierarchy also includes relevant terms of other ontologies. The present ontology has concepts related to dietary assessment tool, dietary assessment questionnaire, dietary data validation, dietary data processing, and dietary data quality descriptions.

### 3.3. Application of ONE

#### 3.3.1. Case Study 1: Study Annotation and Term Query

The annotations for a manuscript [34] and its dataset collected in Cameroon [46] are presented in Table A4 and Table A5 (Appendix A), respectively. Using ONE terms (e.g., “study name”, “study objective”, “study population”, etc.) to link the manuscript/dataset to its meta-data, the manuscript/dataset is annotated according to the data standards and STROBE-nut reporting guidelines included in ONE. Applying ONE avoids confusion when annotating the manuscript and dataset since all term definitions become available. This facilitates the correct understanding by annotators and users of annotated manuscripts and datasets.

#### 3.3.2. Case Study 2: Ontology-Based Inference

Using the annotation in case study 1, the potential for ontology-based inferencing is presented in Table A6 (Appendix A). Using “country”, “study”, and “method” as relationships between the manuscript and its meta-data, the manuscript is annotated as “a cross-sectional study collecting data in Cameroon by 24-h recall method”. The annotation is inferred to a more generic annotation through the taxonomies of terms in the United States National Library of Medicine Medical Subject Headings (MeSH) and ONE ontology. The upper level terms of “MeSH:D002163”, “MeSH:D03430”, and “one:ne00003” are “MeSH:D000350”, “MeSH:D016021”, and “one:ne00001” (second column), respectively. According to the labels of the three upper level terms, the inferred information (third column) is obtained: “this is an epidemiologic study collecting data in Central Africa by dietary assessment method”. The ontology inference now enables integration and a wider search of data. For example, when searching information annotated for “Central Africa”, the present data from “Cameroon” are identifiable. 

#### 3.3.3. Case Study 3: Estimation of Reporting Completeness in a Sample of Nine Manuscripts

The STROBE-nut annotation of nine manuscripts is added under ONE class “case studies: study description” [47]. Table A7a (Appendix A) counts the number of STROBE-nut items described in each manuscript, while Table A7b (Appendix A) reports the frequency of each STROBE-nut item reported in the nine manuscripts. Additional details on the hierarchy of annotation is available in Table A7c (Appendix A). For instance, the study by Mills, Brown, Wrieden, White, and Adams [37] indicates three STROBE-nut items (i.e., Nut-13, Nut-14, and Nut-16) that were reported in the “methods section”, instead of the “results section” of manuscripts as recommended by STROBE-nut.

## 4. Discussion

We reviewed existing ontologies to identify terms for annotating nutritional epidemiologic research output. Ontology terms were collected to describe the minimal information needed to annotate and link research outputs such as data, manuscript, and study protocols to facilitate study identification, retrieval, integration, and reuse. 

To date, an ontology for study level description in nutrition epidemiology is missing. The present work adds value to the Cochrane PICO (i.e., patient, population, or problem; intervention, comparison, and outcome) ontology [5], which is being developed to formulate research questions, and search and characterize clinical studies, as well as meta-analyses. ONE complements the work of GODAN (Global Open Data for Agriculture and Nutrition) [48], LanguaL [16], and FoodEx2 [14] initiatives, which focused on food items and their properties. Moreover, ONE can be considered as an extension of the Epidemiology Ontology (EPO) that summarizes the features of generic epidemiologic studies [26,49].

To our knowledge, it is the first time that an ontology is developed based on manuscript reporting guidelines such as STROBE-nut [50]. Reporting guidelines are widely applied and endorsed by journals as tools to improve completeness of reporting in biomedical research, to enable easier searching, filtering, and navigation of research findings for further policy, practice, or research [51,52]. However, reporting guidelines remain a paper-based initiative. The conversion into a machine-readable representation could expand the use of reporting guidelines to searching and inferring of information. Converting other research reporting guidelines such as CONSORT (CONsolidated Standards of Reporting Trials) [53] or PRISMA (Preferred Reporting Items for Systematic Reviews and Meta-Analyses) [54] into ontologies would significantly improve the scope of their application. For instance, assessment of reporting completeness remains a manual and ad hoc exercise and was only attempted in a handful of cases [55,56,57]. The application of ontologies could potentially be used for automatic monitoring of reporting completeness of manuscripts. It would enable identification of frequently and rarely reported STROBE-nut items and where they are applied in the manuscripts and, as such, provide insights to update the standards [58]. Other potential applications of ontologies for research output include the monitoring of trends in research and identification of neglected areas, as shown in the use of the gene ontology for genetic research [59]. Similar applications are useful for recommendations o minimal data requirements and data quality descriptions.

To update ONE, automated processes will be required [13]. Ontology learning, a process where machines are taught by humans how to build ontologies from text, provides useful prospects in this regard [60]. Ontology learning from text was demonstrated earlier [61]. For instance, Arguello Casteleiro et al. [62] applied deep learning to extract a cardiovascular disease ontology from biomedical literature. However, considerable technical challenges remain, and sustained effort by nutritionists and machine learning expertise will be required. 

Development of user-friendly applications of ontology-based annotation will be required to apply ONE and minimize the burden of ongoing work by researchers. To date, most researchers in nutritional epidemiology are unfamiliar with ontologies. Further ontology development in nutritional epidemiology will, therefore, require the contribution of researchers working in multiple research areas. Additional training and capacity-building efforts are needed to ensure uptake and ownership by the nutrition research community. Ad hoc training sessions were organized previously [63], but will require further development and integration in academic curricula. 

The strength of the current work is the use of existing standards and recommendations that are developed for nutrition research [51,64]. Those standards are developed by and used in the nutrition research community and ensure validity of ONE in the wider research community. Existing ontologies were reviewed as a preparation to convert existing standards into an ontology. As such, the review is a useful resource for researchers and ontology developers in nutritional epidemiology. However, some of the reviewed ontologies did not contain terms that were essential for ONE and consequent ontology-based inferring. 

The existing ontologies reviewed, including ONE, are not yet sufficient to annotate all aspects of nutritional epidemiology. For example, an ontology to connect dietary intake data to food nutrition composition data based on international/local food composition tables is still missing. Meanwhile, ontologies for other reporting guidelines such as CONSORT [53] and PRISMA [54] would facilitate the reporting of findings from other types of research. To enable ontology applications in nutritional epidemiology, additional contributions are required from researchers working on multiple research areas. In addition, four reviewed ontologies (Randomized Controlled Trials Ontology (RCTONT) [65], Non-Randomized Controlled Trials Ontology (NONRCTO) [66], Immune Disorder Ontology (IMMDIS) [67], and Neglected Tropical Disease Ontology (NTDO) [68]) contained errors in the formats and could not be assessed. Identifying these data gaps is hopefully an incentive to address the missing elements.

## 5. Conclusions

To conclude, this study introduced a comprehensive ontology for reporting nutritional epidemiologic studies and data. When applied at scale, application of ONE could enable monitoring of reporting completeness of nutritional epidemiology in the biomedical literature. Ultimately, the generated ontologies should be integrated with other linked data and applied in data collection tools, text editors, journal submission systems, or data repositories for convenient and scalable search, quality checking, etc.

## Figures and Tables

**Figure 1 nutrients-11-01300-f001:**
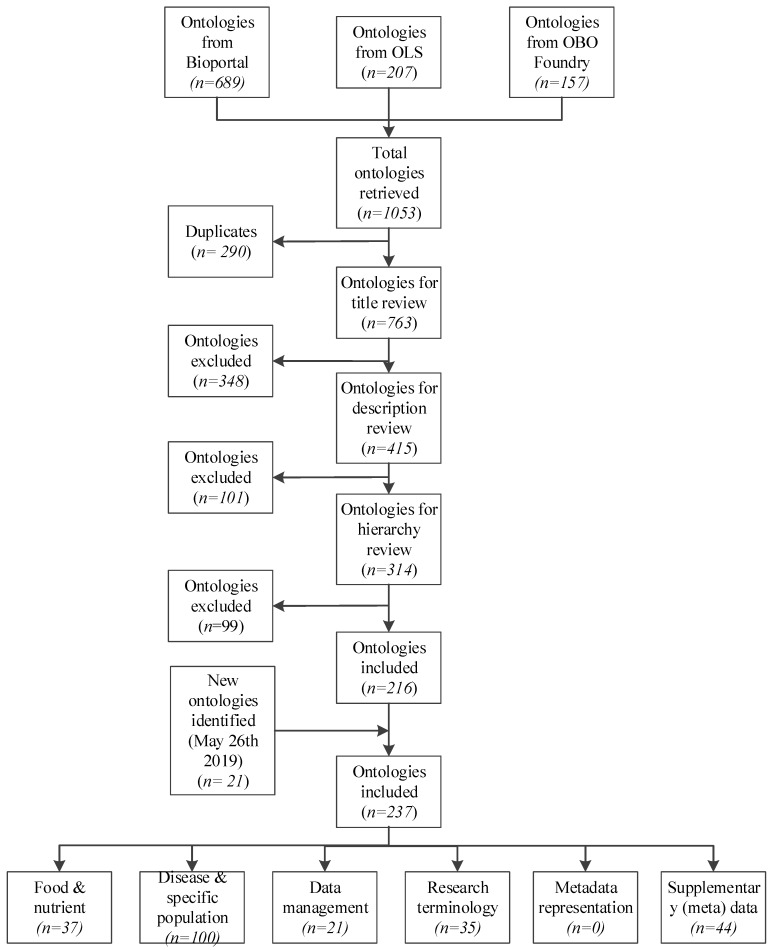
Review and selection process of ontologies for nutritional epidemiology.

**Figure 2 nutrients-11-01300-f002:**
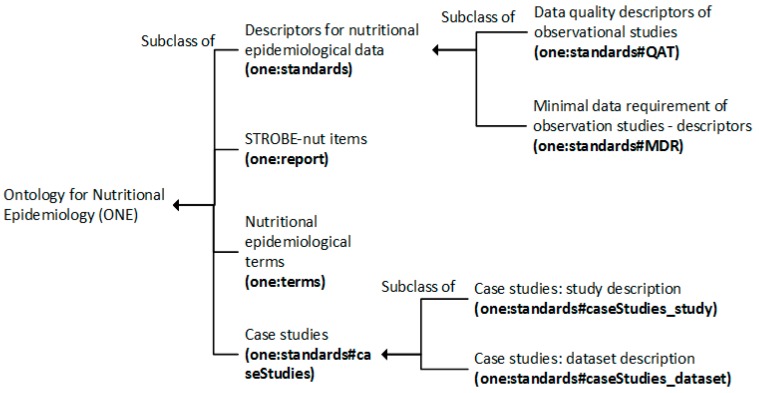
The overall structure of the ontology for nutritional epidemiology (ONE).

**Figure 3 nutrients-11-01300-f003:**
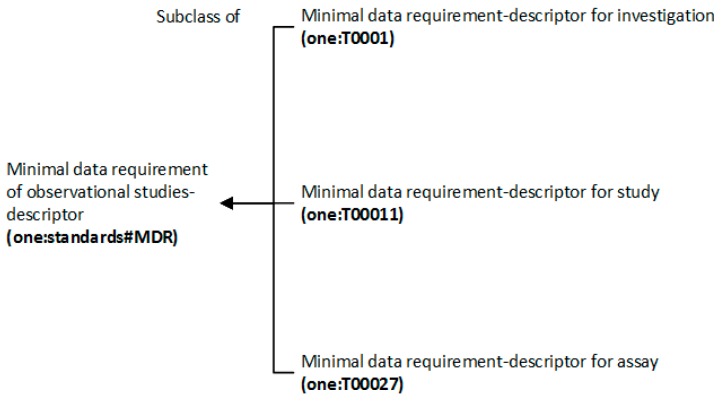
The ontology taxonomy of minimal data requirements.

**Figure 4 nutrients-11-01300-f004:**
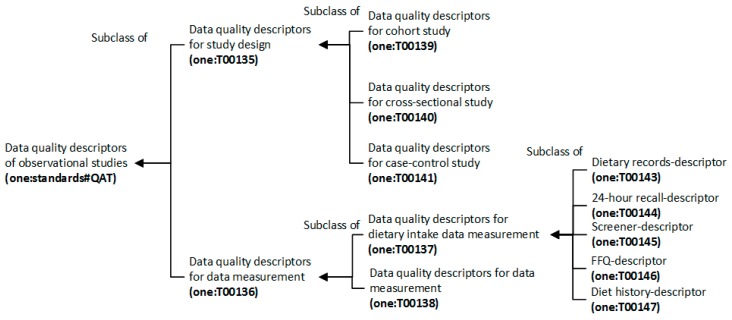
The ontology taxonomy of data quality descriptors of observational studies in nutritional epidemiology.

**Figure 5 nutrients-11-01300-f005:**
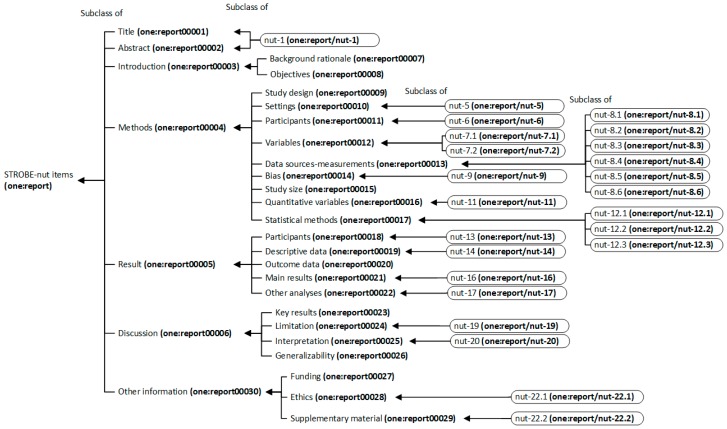
The ontology taxonomy of strengthening the reporting of observational studies in epidemiology (STROBE)-nut (nutritional epidemiology) items.

**Table 1 nutrients-11-01300-t001:** Key concepts used in the manuscript.

Concepts	Descriptions
FAIR [10]	The “findable, accessible, interoperable, and reusable” or FAIR data principles were launched in 2016 to guide data sharing. The FAIR principles are considered key to enhance and enable use of research data.
FoodOn [15]	FoodOn is an ontology to represent knowledge of food in different domains, such as agriculture, medicine, food safety inspection, shopping patterns, sustainable development, etc.
LanguaL and FoodEx2 [16,19]	LanguaL and FoodEx2 are systems for food classification and enable describing, searching, and retrieving data related to food.
MeSH [20]	MeSH stands for “Medical Subject Headings”. It involves hierarchically organized terminology of biomedical information. MeSH is widely applied in National Library of Medicine (NLM) databases for information querying.
NCIT [21]	NCIT stands for the “National Cancer Institute’s Thesaurus”. It involves hierarchically organized terminology/ontology in the cancer domain.
STROBE-nut [4]	As an extension of the STROBE (strengthening the reporting of observational studies in epidemiology) reporting guideline, STROBE-nut (“nut” represents “nutritional epidemiology”) helps researchers to report nutritional epidemiologic research.
RDF [22]	RDF stands for “resource description framework”, and is a standard to describe web resources.

**Table 2 nutrients-11-01300-t002:** Hierarchical structure of nutritional epidemiologic terms.

1st Hierarchy Level	2nd Hierarchy Level	3rd Hierarchy Level
Dietary assessment tool (one:ne00001)	Dietary records (one:ne00002)	Dietary record: short term (one:00042)Dietary record: long term weighted (>7 days) (one:ne00043)Dietary records: PDA (Personal Digital Assistant) technologies (one:ne00007)Dietary records: mobile phone-based technologies (one:ne00008)Dietary records: camera recorder-based technologies (one:ne00009)Dietary records: tape recorder-based technologies (one:ne00010)
24-h recall (one:ne00003)	24-h recall: interactive computer-based technologies (one: 00011)24-h recall: interactive web-based technologies (one: 00012)
Screener (one:ne00004)	Screener: Interactive computer-based technologies (one:ne00013)Screener: Interactive web-based technologies (one:ne00014)Screener: qualitative (only frequency) (one:ne00015)Screener: semi-quantitative (one:ne00016)Screener: quantitative (one:ne00017)
Food Frequency Questionnaire (FFQ) (one:ne00005)	FFQ: interactive computer-based technologies (one:ne00018)FFQ: interactive web-based technologies (one:ne00019)FFQ: qualitative (only frequency) (one:ne00020)FFQ: semi-quantitative (one:ne00021)FFQ: quantitative (one:ne00022)
Diet history (one:ne00006)	
Dietary intake data (one:ne00023)	Unadjusted data (preferred option) (one:ne00024)Adjusted data for total energy intake using density method (one:ne00025)Adjusted data for total energy intake using residual method (one:ne00026)Estimates of usual intake from short-term measurements (one:ne00027)	
(External upper level: administration (NCIT:C25409))Dietary assessment administration (one:ne00028)	Proxy-administered (one:ne00029)Self-administered not verified by interviewer (one:ne00030)Self-administered and checked by interviewer (one:ne00031)Interview-administered (one:ne00032)Interview-administered using AMPM (Automated Multiple Pass Method) (one:ne00033)	
(External upper level: questionnaire (NCIT_C17048))Dietary assessment questionnaire (one:ne00034)	Self-developed questionnaires (one:ne00035)Use of standardized questionnaire (one:ne00036)Adopted other questionnaires (one:ne00037)	
(External upper level: content validity (NCIT_C78690))Content validity of dietary assessment questionnaire (one:ne00038)	Verified content validity in another population (one:ne00039)Verified content validity in a comparable population in terms of both age and dietary habits (one:ne00040)	
Reference of dietary assessment questionnaire validation (one:ne00041)	Dietary assessment methods (one:ne00001)	
Objective methods (one:ne00044)	Biomarker of dietary intake (one:ne00045)
Validated information (OBI_0302838)Validated information of dietary assessment questionnaire (one:ne00046)	Properties of dietary assessment questionnaire (one:ne00047)	Inter-rater reliability (NCIT_C78688)
Frequency options to identify between-person variations (one:ne00048)	
Food items lead to underestimated target nutrients intake (one:ne00049)	
Validation type for dietary assessment questionnaire (one:ne00050)	Concurrent validity (OBCS_0000160) precision (NCIT_C48045)	
Quantification of portion sizes (one:ne00051)	Not quantified (one:ne00052)Standard portion sizes without aids (one:ne00053)Standard portion sizes with aids (one:ne00054)Portion sizes are assessed digitally but not verified by trained staff (one:ne00055)Portion sizes are assessed digitally and verified by trained staff (or packaging) (one:ne00056)	
Portion size of dietary intake data (one:ne00057)	Directly expressed portion size (one:ne00058)Converted portion size (one:ne00059)Unconverted portion size (one:ne00060)	
Matched consumed food to referred food composition data (one:ne00060)	Exact matching (one:ne00061)Matched to means of min. 3 food items (one:ne00062)Matched to same food items with similar moisture content (one:ne00063)Matched to a different food (one:ne00064)Percentage in xsd:decimal	
Representativeness of the week/weekend days (one:ne00065)	Weekend (NCIT_C137684)Weekday (NCIT_C86936)	
Number of recall/measurement days per individual (one:ne00066)	xsd:integer	
Selection of recall/measurement days (one:ne00067)	Convenience selection (one:ne00068)Consecutive days (one:ne00069)Non-consecutive, non-random days (one:ne00070)Randomly over the week (one:ne00071)	
The time of diet records (one:ne00072)	Not during eating occasions nor immediately after (one:ne00073)Immediately after eating occasion (one:ne00074)During eating occasion (one:ne00075)	
Food quantification method (one:ne00076)	Food quantification method tailored to the characteristics of the population (one:ne00077)Food quantification method not specifically tailored to the characteristics of the population (one:ne00078)

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
