# Peer review of "An Ontology to Standardize Research Output of Nutritional Epidemiology: From Paper-Based Standards to Linked Content"

_nutrients, 2019, doi:10.3390/nu11061300_

Round 1

Reviewer 1 Report

This article proposes an ontology for nutritional epidemiology. The study is timely and needed to advance nutritional epidemiological research. However, I have a few suggestions that may help to improve the paper further:
1. The review of existing ontologies was done by April 13th 2018. It’s a year ago. Updated searches and reviews should be conducted to make the study current.

2. The ontology proposed includes reporting guidelines. However, the guidelines are only for observational studies. It would be useful to the field if the guidelines for RCTs could be proposed, especially, as the authors mentioned, the existing RCT ontologies contain errors in the formats.

3. Could the reporting guidelines for observational studies  be grouped/stratified by study design, eg. cross-sectional studies, prospective/retrospective cohort  studies, case-control studies, etc.? These guidelines are lacking and needed to guide more meaningful reporting.

4. Section 3.2.3 needs improvement. The Hierarchical levels needs to be better explained and justified. For example, Dietary assessment methods and Dietary assessment questionnaire are both at level 1, while questionnaire is just one assessment method. Also, FFQ is one type of assessment questionnaire. The reasoning behind the hierarchical structure should be further clarified.

4. It is unclear how the new ontology could help facilitate “Integrated analysis of shared data”, and “retrieval and use of shared nutritional epidemiologic data”. A case study could be added to illustrate the application.

Author Response

Response to Reviewer 1 Comments

Point 1: The review of existing ontologies was done by April 13th 2018. It’s a year ago. Updated searches and reviews should be conducted to make the study current.

Response 1:

Thank you for the comment. An update of the review has been carried out on 26th May, 2019, and the main changes of the review result are summarized below:

1) The review of ontologies in the three ontology libraries was updated on 26th May, 2019. We identified 93 new ontologies published after April 13th, 2019 (12 ontologies in OBO foundry, 18 ontologies in OLS and 63 ontologies in BioPortal). After removing duplicates and reviewing, 21 of the retrieved ontologies were selected. The 21 ontologies include 14 ontologies to describe “disease & special population”, 4 ontologies to describe “food and nutrients”, and 3 ontologies to describe “research terminology”;

2) We re-assessed the 5 inaccessible ontologies. Four of them are still inaccessible. However, the Ontology of Physical Exercises (OPE, https://bioportal.bioontology.org/ontologies/OPE) had been corrected by developers and it is accessible now.

3) The relevant statistics have been updated. In the manuscript, the relevant paragraphs and the statistics showed in “Figure 1”, “Figure A1” and “Table A2” have been updated.

Point 2: The ontology proposed includes reporting guidelines. However, the guidelines are only for observational studies. It would be useful to the field if the guidelines for RCTs could be proposed, especially, as the authors mentioned, the existing RCT ontologies contain errors in the formats.

Response 2:

We agree that ontologies for RCT would be a relevant and timely contribution. The present work was led by the research group developing the extension of STROBE reporting guideline for nutritional epidemiologic research (e.g. STROBE-nut). In this sense, we feel qualified to develop and propose a taxonomic hierarchy of the concepts presented in the present manuscript.

We therefore feel it is best that the CONSORT group leads the development of the corresponding ontology for reporting RCTs. Our work illustrates the potential to apply reporting guidelines as a basis for ontology and paves the way for further work in this area. In this sense, will reach out to CONSORT for technical support to facilitate the development of their ontology.

As we stated in the second last paragraph of “Discussion”: “The reviewed existing ontologies, including ONE, are not yet able to annotate all aspects in nutritional epidemiology… To enable ontology applications in nutritional epidemiology, contributions are still required from researchers working on multiple research areas.”

To clarify your comment, an extra sentence about CONSORT/RCT has been added in the paragraph (line 291-292): “Converting other research reporting guidelines such as CONSORT [1] or PRISMA [2] into ontologies, would significantly improve the scope of their application.”

Point 3: Could the reporting guidelines for observational studies be grouped/stratified by study design, e.g. cross-sectional studies, prospective/retrospective cohort  studies, case-control studies, etc.? These guidelines are lacking and needed to guide more meaningful reporting.

Response 3:

Thank you for the comments. The grouping of items by study designs was done according to the structure of the STROBE and STROBE-nut reporting guidelines, which combines the different observational studies. The STROBE items refer to the following in this regard:

Eighteen items apply to all three study designs whereas four are design-specific. Starred items (for example item 8*) indicate that the information should be given separately for cases and controls in case-control studies, or exposed and unexposed groups in cohort and cross-sectional studies.”

(Reference. Vandenbroucke JP, von Elm E, Altman DG, Gøtzsche PC, Mulrow CD, Pocock SJ, et al. (2007) Strengthening the Reporting of Observational Studies in Epidemiology (STROBE): Explanation and Elaboration. PLoS Med 4(10): e297. https://doi.org/10.1371/journal.pmed.0040297)

The general purpose of STROBE-nut is to help researchers report specific characteristics for the different nutritional epidemiological studies. As a result, grouping reporting items according to study design was not considered when developing the extension for reporting nutritional epidemiologic characteristics.

Point 4: Section 3.2.3 needs improvement. The hierarchical levels needs to be better explained and justified. For example, Dietary assessment methods and Dietary assessment questionnaire are both at level 1, while questionnaire is just one assessment method. Also, FFQ is one type of assessment questionnaire. The reasoning behind the hierarchical structure should be further clarified.

Response 4:

Thank you for the comment. The relevant content has been edited (line 235 – line 239):

“All the terms were arranged according to the relevant descriptors listed in the minimal data requirements [3] and data quality descriptors [4]. The terms at the first hierarchy level are the descriptors (i.e. nutritional epidemiologic terms), while the terms at the second and third hierarchy levels are the options of descriptors (i.e. terms with more specific descriptions used for specific conditions).”

Meanwhile, following your comment “FFQ is one type of assessment questionnaire”, we renamed the category as “Dietary assessment tool (one:ne00001)”.

Point 5: It is unclear how the new ontology could help facilitate “Integrated analysis of shared data”, and “retrieval and use of shared nutritional epidemiologic data”. A case study could be added to illustrate the application.

Response 5:

We did consider integrating such case studies in the manuscript. The case studies suggested however, require two Semantic Web technologies, SPARQL query language (https://www.w3.org/TR/rdf-sparql-query/) for data retrieval and data format exchange, and SWRL/SQWRL rule language (https://www.w3.org/Submission/2004/SUBM-SWRL-20040521/) for data reasoning. Integrating these elements in this manuscript would be challenging and outside of its initial scope. We therefore consider it more appropriate to describe such case studies in a separate manuscript which gives us additional space to describe SPARQL, SWRL as well as their applications in more detail. In fact, currently, we are writing a new article to showcase the application of ONE based on the SPARQL and SWRL technologies applications for integrated analysis of nutritional epidemiologic data.

References

[1] Moher D, Hopewell S, Schulz KF, Montori V, Gotzsche PC, Devereaux PJ, et al. CONSORT 2010 explanation and elaboration: updated guidelines for reporting parallel group randomised trials. BMJ. 2010;340:c869.

[2] Moher D, Liberati A, Tetzlaff J, Altman DG, Group P. Preferred reporting items for systematic reviews and meta-analyses: the PRISMA statement. PLoS Med. 2009;6:e1000097.

[3] Pinart M, Nimptsch K, Bouwman J, Dragsted LO, Yang C, De Cock N, et al. Joint Data Analysis in Nutritional Epidemiology: Identification of Observational Studies and Minimal Requirements. J Nutr. 2018;148:285-97.

[4] Yang C, Pinart M, Kolsteren P, Van Camp J, De Cock N, Nimptsch K, et al. Perspective: Essential Study Quality Descriptors for Data from Nutritional Epidemiologic Research. Adv Nutr. 2017;8:639-51.

Reviewer 2 Report

"progress towards an ontology for nutritional epidemiology is poor" i line 62, it's probably too hard, maybe it would be better - progress is not that we'd expect

Congrtulations for authors it is very good idea, the use of the system may cause some difficulties, but its users should help in making changes that will make them more accessible

Author Response

Response to Reviewer 2 Comments

Point 1: progress towards an ontology for nutritional epidemiology is poor" i line 62, it's probably too hard, maybe it would be better - progress is not that we'd expect.

Response 1:

Thank you for the comment. The sentence has been edited.

Line 62: “…progress towards an ontology for nutritional epidemiology is limited.”

Point 2: Congrtulations for authors it is very good idea, the use of the system may cause some difficulties, but its users should help in making changes that will make them more accessible

Response 2:

We appreciate the positive feedback and agree with the statement.  Introducing an ontology for nutritional epidemiologists is really challenging but has potential to add considerable value. Ontology/Semantic Web technologies can bring nutrition epidemiology to another level. Like the benefits brought by GO (Gene Ontology) to the field of Gene Science, we believe that ONE, as well as its applications could facilitate nutritional epidemiologic research step by step (e.g. save time and cost for data management, and generate new research findings through data-based semi-automatic inference and machine learning).

Reviewer 3 Report

Dear Authors

After reading your paper, I would have 3 areas of concern: 1. Integration with other ontologies, 2 definition of nutritional epidemiology, 3. Use of acronyms

INTEGRATION

It is difficult to imagine food and nutrition as stand-alone determinants of health (except in very specific cases like inborn errors of metabolism).  Other lifestyle factors such as exercise and sleep, as well as disease (NCDs or not) are also important.  It is therefore important to be able to access multidisciplinary epidemiological evidence. 

Therefore, I would like to know whether, and if so how, ONE will be integrated into a wider network of epidemiological ontologies. 

For example, it would be pertinent if you could  explain how your work complements (or not) the Epidemiology Ontology that was developed under the EU-funded EPIWORK project.  This could be discussed in light of Pesquita et al’s paper

Pesquita C, Ferreira JD, Couto FM, Silva MJ. The epidemiology ontology: an ontology for the semantic annotation of epidemiological resources. J Biomed Semantics. 2014;5(1):4. Published 2014 Jan 17. doi:10.1186/2041-1480-5-4

As well as

Ferreira JD, Paolotti D, Couto FM, Silva MJ. On the usefulness of ontologies in epidemiology research and practice. J Epidemiol Community Health. 2013;67(5):385–388. doi:10.1136/jech-2012-201142

Will ONE be integrated into NERO (Network of Epidemiology Related Ontologies) – if not, why not?

DEFINITION

Line 88 – You say, “Nutritional epidemiology is an interdisciplinary science that builds on other disciplines such as nutrition, food science, medicine and epidemiology.”  I think your definition of nutritional epidemiology is confusing and should be rephrased.

Nutritional epidemiology looks at the relationship between nutrition and health but is part of the broader field  of epidemiology – as stated by Boeing (2013) who says, “Nutritional epidemiology is a subdiscipline of epidemiology and provides specific knowledge to nutritional science.” Eur J Clin Nutr. 2013 May;67(5):424-9. doi: 10.1038/ejcn.2013.47. Epub 2013 Feb 27. 

ACRONYMS

I think your paper is quite technical (difficult to read) if you are not working in this field.  This is exacerbated by the large number of acronyms.  I would appreciate seeing a table of acronyms with simple explanations.

Author Response

Response to Reviewer 3 Comments

Point 1: INTEGRATION

It is difficult to imagine food and nutrition as stand-alone determinants of health (except in very specific cases like inborn errors of metabolism).  Other lifestyle factors such as exercise and sleep, as well as disease (NCDs or not) are also important.  It is therefore important to be able to access multidisciplinary epidemiological evidence.

Therefore, I would like to know whether, and if so how, ONE will be integrated into a wider network of epidemiological ontologies.

For example, it would be pertinent if you could  explain how your work complements (or not) the Epidemiology Ontology that was developed under the EU-funded EPIWORK project.  This could be discussed in light of Pesquita et al’s paper

Pesquita C, Ferreira JD, Couto FM, Silva MJ. The epidemiology ontology: an ontology for the semantic annotation of epidemiological resources. J Biomed Semantics. 2014;5(1):4. Published 2014 Jan 17. doi:10.1186/2041-1480-5-4

As well as

Ferreira JD, Paolotti D, Couto FM, Silva MJ. On the usefulness of ontologies in epidemiology research and practice. J Epidemiol Community Health. 2013;67(5):385–388. doi:10.1136/jech-2012-201142

Will ONE be integrated into NERO (Network of Epidemiology Related Ontologies) – if not, why not?

Response 1:

Thank you for the suggestions. We are keen to build bridges and explore possibilities to integrate our ontology into a larger initiative. We reviewed the two suggested papers. EPO is indeed a nice ontology resource and useful to describe generic features of epidemiologic studies. We therefore add a few sentences to discuss the relation between ONE and EPO (line 281-284):

“ONE complements the work of GODAN [1], LanguaL [2] and FoodEx2 [3] initiatives, which have focused on food items and their properties. Moreover, ONE can be considered as an extension of the Epidemiology Ontology” (EPO) that summarizes the features of generic epidemiologic studies [4, 5].”

We were eager to link up with the NERO initiative and consider the possibility to use ONE as an extension of EPO. Unfortunately however, NERO project was terminated in 2013. EPO/NERO is currently deprecated in OBO foundry (http://www.obofoundry.org/ontology/epo.html), and its namespace “EPO” has been occupied by another ontology named “Early Pregnancy Ontology” (https://bioportal.bioontology.org/ontologies/EPO).

Point 2: DEFINITION

Line 88 – You say, “Nutritional epidemiology is an interdisciplinary science that builds on other disciplines such as nutrition, food science, medicine and epidemiology.”  I think your definition of nutritional epidemiology is confusing and should be rephrased.

Nutritional epidemiology looks at the relationship between nutrition and health but is part of the broader field  of epidemiology – as stated by Boeing (2013) who says, “Nutritional epidemiology is a subdiscipline of epidemiology and provides specific knowledge to nutritional science.” Eur J Clin Nutr. 2013 May;67(5):424-9. doi: 10.1038/ejcn.2013.47. Epub 2013 Feb 27.  

Response 2:

Thank you for your comment. We edited the sentence in the manuscript.

Line 95-99: “As a sub-discipline of epidemiology, nutritional epidemiology analyses the relationship between dietary intake and health [6]. As an interdisciplinary science, nutritional epidemiology also requires knowledge from other disciplines such as nutrition, food science, medicine, etc. Instead of developing a new stand-alone ontology, we therefore first considered existing ontologies in epidemiology [4] as well as the relevant disciplines, and then identified missing elements for nutritional epidemiology [7].”

Point 2: ACRONYMS

 I think your paper is quite technical (difficult to read) if you are not working in this field.  This is exacerbated by the large number of acronyms.  I would appreciate seeing a table of acronyms with simple explanations.

Response 3:

Thanks. It’s an important reminder. A table of acronyms has been added, and indicated in the introduction (below).

Concepts

Descriptions

FAIR [8]

The “Findable, Accessible, Interoperable and Reusable” or   FAIR data principles were launched in 2016 to guide data sharing. The FAIR   principles are considered key to enhance and enable use of research data.

FoodOn [9]

FoodOn is an ontology to represent knowledge of food in   different domains, such as agriculture, medicine, food safety inspection,   shopping patterns, sustainable development, etc.

LanguaL and FoodEx2 [2, 10]

LanguaL and FoodEx2 are systems for food classification   and enable describing, searching and retrieving data related to food.

MeSH [11]

MeSH stands for “Medical Subject Headings”. It is a   hierarchically-organized terminology of biomedical information. MeSH is   widely applied in NLM databases for information querying.

NCIT [12]

NCIT stands for “The National Cancer Institute's   Thesaurus”. It is a hierarchically-organized terminology/ontology in the   cancer domain.

STROBE-nut [13]

As an extension of STROBE (STrengthening the Reporting of   OBservational studies in Epidemiology) reporting guideline, STROBE-nut (“nut”   represents “nutritional epidemiology”) helps researchers to report   nutritional epidemiologic research.

RDF [14]

RDF stands for “Resource Description Framework”, and is a   standard to describe web resources.

References

[1] GODAN. Global Open Data for Agriculture and Nutrition. 2017.

[2] Danish Food Informatics. LanguaL - the International Framework for Food Description. 2015.

[3] European Food Safety Authority. The food classification and description system FoodEx2

[4] Pesquita C, Ferreira JD, Couto FM, Silva MJ. The epidemiology ontology: an ontology for the semantic annotation of epidemiological resources. J Biomed Semantics. 2014;5:4.

[5] Ferreira JD, Paolotti D, Couto FM, Silva MJ. On the usefulness of ontologies in epidemiology research and practice. J Epidemiol Community Health. 2013;67:385-8.

[6] Boeing H. Nutritional epidemiology: New perspectives for understanding the diet-disease relationship? Eur J Clin Nutr. 2013;67:424-9.

[7] Lemay DG, Zivkovic AM, German JB. Building the bridges to bioinformatics in nutrition research. Am J Clin Nutr. 2007;86:1261-9.

[8] Wilkinson MD, Dumontier M, Aalbersberg IJ, Appleton G, Axton M, Baak A, et al. The FAIR Guiding Principles for scientific data management and stewardship. Sci Data. 2016;3:160018.

[9] Dooley MD, Griffiths JE, Gosal SG, Buttigieg LP, Hoehndorf R, Lange CM, et al. FoodOn: a harmonized food ontology to increase global food traceability, quality control and data integration. npj science of food. 2018;2.

[10] Eftimov T, Korosec P, Korousic Seljak B. StandFood: Standardization of Foods Using a Semi-Automatic System for Classifying and Describing Foods According to FoodEx2. Nutrients. 2017;9.

[11] National Library of Medicine. Medical Subject Headings 2017. NLM; 2016.

[12] Golbeck J, Fragoso G, Hartel F, Hendler J, Oberthaler J, Parsia B. The National Cancer Institute's Thesaurus and Ontology. SSRN Electronic Journal. 2003.

[13] Lachat C, Hawwash D, Ocké MC, Berg C, Forsum E, Hörnell A, et al. STrengthening the Reporting of OBservational studies in Epidemiology – Nutritional Epidemiology (STROBE-nut): an extension of the STROBE statement. Plos Medicine. 2016.

[14] Brickley D, Guha RV. RDF Schema 1.1. W3C Recommendation; 2014.

Round 2

Reviewer 1 Report

Accept in present form